# Efficacy of a Turkey Herpesvirus Vectored Newcastle Disease Vaccine against Genotype VII.1.1 Virus: Challenge Route Affects Shedding Pattern

**DOI:** 10.3390/vaccines9010037

**Published:** 2021-01-11

**Authors:** Vilmos Palya, Tímea Tatár-Kis, Abdel Satar A. Arafa, Balázs Felföldi, Tamás Mató, Ahmed Setta

**Affiliations:** 1Scientific Support and Investigation Unit, Ceva-Phylaxia Co. Ltd., Ceva Animal Health, 5 Szallas Utca, 1107 Budapest, Hungary; timea.tatar@ceva.com (T.T.-K.); balazs.felfoldi@ceva.com (B.F.); tamas.mato@ceva.com (T.M.); 2Central Laboratory for Quality Control of Poultry Production, Animal Health Research Institute, Agriculture Research Center, 7 Nadi El-Seid Street, Dokki, Giza 12618, Egypt; abd.arafa@gmail.com; 3Ceva Animal Health, 5 Street 277, Maadi, Cairo 11435, Egypt; ahmed.setta@ceva.com

**Keywords:** Newcastle disease virus, vaccine, recombinant turkey herpesvirus, genotype VII, challenge route, shedding

## Abstract

The control of Newcastle disease (ND) highly relies on vaccination. Immunity provided by a ND vaccine can be characterized by measuring the level of clinical protection and reduction in challenge virus shedding. The extent of shedding depends a lot on the characteristics of vaccine used and the quality of vaccination, but influenced also by the genotype of the challenge virus. We demonstrated that vaccination of SPF chicks with recombinant herpesvirus of turkey expressing the F-gene of genotype I ND virus (rHVT-ND) provided complete clinical protection against heterologous genotype VII.1.1 ND virus strain and reduced challenge virus shedding significantly. 100% of clinical protection was achieved already by 3 weeks of age, irrespective of the challenge route (intra-muscular or intra-nasal) and vaccination blocked cloacal shedding almost completely. Interestingly, oro-nasal shedding was different in the two challenge routes: less efficiently controlled following intra-nasal than intra-muscular challenge. Differences in the shedding pattern between the two challenge routes indicate that rHVT-ND vaccine induces strong systemic immunity, that is capable to control challenge virus dissemination in the body (no cloacal shedding), even when it is a heterologous strain, but less efficiently, although highly significantly (*p* < 0.001) suppresses the local replication of the challenge virus in the upper respiratory mucosa and consequent oro-nasal shedding.

## 1. Introduction

Among viral diseases affecting poultry production Newcastle disease (ND) is one of the most common and serious one [1]. It is caused by viruses of genus *Orthoavulavirus*, *Avian orthoavulavirus 1* species [2] formerly designated as *Avian avulavirus 1*, that are commonly known as *Avian paramyxovirus 1* (APMV 1) or Newcastle disease viruses (NDV). The virulent forms of NDV cause a devastating disease of poultry throughout Asia, Africa, the Middle East, and Central and South America even today [3,4]. In the past decades, there has been a major shift in virulence among the velogenic NDV (vNDV) strains that have been identified as prevalent in poultry [5,6].

Based on complete genome sequences, NDV isolates were separated into two classes [7]. Further analysis of the full-length fusion (F) gene sequences defined 21 genotypes of Class II viruses so far [2]. Viruses sharing temporal, geographical or epidemiological parameters tend to fall into specific genotypes [3,8]. Today, vNDV strains associated with outbreaks in commercial poultry farms and backyard chickens in the Middle East and North Africa predominantly belong to the genotype VII.1.1 of NDV [3,8,9,10,11,12].

Control of Newcastle disease (ND), in addition to good biosecurity practices, primarily relies on preventive vaccination of flocks and culling of infected and at risk of being infected birds [13,14]. Despite the disease being caused by one single serotype of the ND virus, and thus, by definition, any NDV strain utilized to prepare a vaccine should induce protection against morbidity and mortality from any vNDV infection, the control of ND still remains difficult [13,15,16]. In endemic countries, most vaccination programs for ND include the use of both live (lentogenic NDV strains) and inactivated (killed) vaccines in order to induce a good protective immunity while producing minimal adverse effects in the birds [13,17]. Both types of vaccine have their advantages and disadvantages, but the continuous outbreaks of fatal ND in commercial poultry flocks in many parts of the world indicate that routine vaccination in the field often fails to induce sufficiently high levels of immunity to control ND [13,18,19]. The reasons for this are multiple. Apart from poor vaccination techniques and mismanagement, the presence of maternally derived antibodies (MDA) also interferes with the establishment of an early and persisting immunity after single or even repeated vaccination during the first 2–3 weeks of life [20].

Therefore, the need for the availability of more efficacious and safe ND vaccine is ever increasing. Furthermore, there is an increasing demand for better vaccination strategies so as to promote a more efficient and accurate means of delivering a uniform vaccine dose to each individual chickens. A promising approach to achieve the above goals is the use of vector vaccines [21]. Herpesvirus of turkeys (HVT) and attenuated Marek’s disease (MDV 1) viruses expressing foreign antigens related to poultry pathogens are considered as the most potent vectors [21].The envelope of NDV contains two transmembrane proteins, the haemagglutinin-neuraminidase (HN) and the fusion (F) proteins which are important for virus infectivity and pathogenicity. The HN protein is responsible for the attachment of virus to the host cells, while the F protein mediates fusion of the virion envelop with the cell membrane [22]. Both of them are considered as the major protective antigens since they induce virus neutralizing antibodies, although HN was shown to induce lower neutralizing antibody titres and lower protection. The results of experiments with recombinant turkey herpesvirus expressing either the HN or the F proteins suggest that they are independent neutralization and protective antigens, and the F protein provide better protection than the HN protein [23]. HVT as vector, which contains and expresses protective antigen of NDV, that is typically the F glycoprotein has been shown to elicit immune response and provide protection against lethal challenge with velogenic strains of NDV [24,25,26,27]. As in case of HVT itself, long term virus persistence was shown for an HVT-vectored ND (rHVT-ND) vaccine also [26]. Additionally, rHVT-ND construct appeared to be less sensitive to interference with MDA, than the conventional ND vaccines, which adds further useful characteristic to this vector vaccine [25,26,27,28].

The study reported here was designed to evaluate the efficacy of a rHVT-ND vaccine expressing the F protein of NDV, against challenge with a genotype VII.1.1 (formerly VIId) velogenic NDV strain using both the intra-nasal (mimicking the natural route of infection) and the intra-muscular routes. Criteria for efficacy evaluation were extended to include the analyses of the challenge virus excretion patterns in the two groups of chickens infected by different routes.

## 2. Materials and Methods

### 2.1. Animals

For this study, 132 day-old SPF chickens purchased from a local supplier (Biovo Kft, Mohács, Hungary) were used. They were randomly allocated into eight treatment groups that were accommodated separately in Biosafety Level 3 animal rooms at Prophyl Kft., Mohács, Hungary. They were fed adequate for the age of the birds, and water was supplied ad libitum.

The study has been conducted in compliance with the provisions of Directive 2010/63/EU, Hungarian Act No. XXVIII/1998, and the Hungarian Governmental Decree No. 40/2013. (II.14.) on the protection of animals used for scientific purposes. The animal study was undertaken after approval of the Hungarian competent authorities (permission no.: BAI/35/56-92/2017). The execution of the study was under the supervision of qualified veterinarians. Chickens showing the terminal stage of the disease were removed, euthanized and counted as mortality.

### 2.2. Vaccine and Vaccination

A total of 92 SPF chicks were vaccinated at day old with the rHVT-ND vaccine (Vectormune^®^ ND, Ceva Sante Animale, Libourne, France), which contains F gene derived from a class II genotype 1 NDV strain (Appendix A, GenBank accession number: M24692). The vaccine was reconstituted in its specific diluent to contain a dose of 1500 plaque forming units/chick in 0.2 mL and was injected by the subcutaneous route under the skin of the neck. 40 hatch-mate chicks were kept as unvaccinated controls. Vaccine-take (presence of vaccine virus in vaccinated birds) was followed by detection of rHVT-ND from feather tips with pulp collected from individual birds at 7, 15, 20, and 43 days of age and spleen samples collected at 20, 36, and 43 days of age.

### 2.3. Serology

To monitor antibody response to vaccination, blood samples were collected at 7, 15, 20, 28, 35, and 43 days of age. The antibody level against NDV was measured by a commercial ELISA kit that is suited for monitoring the immune response to rHVT-ND vaccine (ID Screen^®^ Newcastle Disease Indirect ELISA Kit, Product code: NDVS, IDVet, Montpellier, France) following the manufacturer’s instruction and by haemagglutination inhibition (HI) test (LaSota antigen, 4 haemagglutinating units) according to the OIE Terrestrial Manual [29]. The positivity limits for the ELISA and HI tests were above 993 ELISA titre and above 2 log_2_ HI titre, respectively. Post-challenge samples (samples collected 14 days after the 1st and 2nd challenge date and samples of non-challenged chickens collected at termination of 2nd challenge) were also tested with IDScreen^®^ Newcastle Disease Nucleoprotein Indirect ELISA (code: NDVNP, IDVet, Montpellier, France).

### 2.4. Challenge Virus

An Egyptian velogenic viscerotropic NDV strain (NDV-B7-RLQP-CH-EG-12) isolated from a 22 days old broiler chicken and provided by NLQP, Egypt, was used in the experiment (designation used in phylogenetic tree is D3538). Phylogenetic analysis of this strain was performed based on the full F gene nucleotide sequence. Reference sequences of velogenic NDV strains were obtained from the GenBank (accession numbers are shown in the phylogenetic tree, Appendix A). Comparison of nucleotide sequences was performed with Kimura’s two-parameters method, and the phylogenetic tree was constructed by the neighbour-joining method using MEGA 7 software [30]. The positions of this challenge strain and the strain providing the F gene insert of the rHVT-ND vaccine are highlighted in the phylogenetic tree (Appendix A). The sequence of the challenge virus strain is deposited in the GenBank (accession number: MT876631).

### 2.5. Challenge Infections and Post-Challenge Samplings

Challenges were performed at 20 and 28 days of age in separate subgroups of chickens. We used two challenge routes on both occasions in parallel to compare the efficacy of vaccine when the challenge virus was administered either by the intra-muscular (official method of European Pharmacopoeia) or by the intra-nasal route (the latter method is to mimic natural infection). Challenge was done with a dose of 5.0 log_10_ ELD_50_/chicken (median embryo lethal dose) regardless the age of birds and challenge route used. At both dates, with both challenge routes, 20 vaccinated and 10 non-vaccinated chickens were submitted to challenge infection. A summary of study set-up is given in Table 1.

Post-challenge monitoring included daily clinical observations for two weeks. All birds that died or showed clinical signs indicative of ND were recorded as non-protected. All vaccinated birds showing ND specific clinical signs, and the ones surviving the intramuscular challenge at 3 weeks of age, were sampled for rHVT-ND detection from the spleen to check for the presence of vaccine virus.

To monitor challenge virus shedding, oro-nasal and cloacal swabs were collected at four days post-challenge (dpch) from 10 chickens per group (first 10 serial numbers out of the 20 challenged chickens in the vaccinated groups were pre-selected randomly for testing of virus shedding, while all of the 10 birds in the control groups were sampled).

At the termination of both challenges (14 days after the 1st and 2nd challenge date) blood samples were collected and tested with IDScreen^®^ Newcastle Disease Nucleoprotein Indirect ELISA (code: NDVNP, IDVet, Montpellier, France). Non-challenged vaccinated chickens (*n* = 12) were also blood sampled at the termination of both challenges to serve as controls for monitoring the booster effect of the challenge on the humoral antibody level of vaccinated chickens. These birds served also to check the presence of the rHVT-ND vaccine in the spleen and feather pulp at the termination of the 2nd challenge experiment (43 days of age).

### 2.6. ND Challenge Virus Shedding Measurement

Challenge virus amount was quantified from the swabs using a quantitative reverse-transcription real-time PCR (qRT-PCR) that amplified a fragment of the matrix gene (TaqMan^®^ NDV reagents and controls, one-step qRT-PCR for NDV RNA, Applied Biosystem, Foster City, CA, USA). The PCR primers and probe were designed by Wise et al. [31]. The positivity limit was set at a Ct of 36 or below, according to the manufacturer’s recommendations. The quantity of NDV in the samples was analysed by comparison of Ct values, since there was no enough data on the correlation of live NDV titre and Ct values in the 4 dpch swab samples to calculate the live virus content of the samples.

### 2.7. Statistical Analysis

The amount of ND challenge virus shed was compared between the vaccinated groups challenged at different ages and by different routes, and between the vaccinated and control groups challenged at the same age and via the same route. Ct values of oro-nasal and cloacal swabs were evaluated separately by using Mann–Whitney test at 95% confidence level (*p* < 0.05).

The effect of the challenge on the humoral antibody level was analysed by comparing the log_2_ HI titres or ID Screen^®^ Newcastle Disease Indirect ELISA titres obtained in the vaccinated and challenged subgroups with the corresponding non-challenged vaccinated groupmates sampled at the same age. Evaluation was performed with Mann–Whitney test at 95% confidence level (*p* < 0.05).

## 3. Results

### 3.1. Vaccine-Take Detection and Humoral Immune Response to Vaccination

Vaccine virus was detectable as soon as at seven days of age in the vaccinated chickens (Table 2). Feather tip samples resulted in 90% positivity at seven days of age, which increased to 100% by 15 days of age. Similarly 100% positivity was found in the spleen samples collected at three, five, and six weeks of age. The rate of rHVT-ND detection decreased only slightly with age in the spleen samples, while there was a marked decrease after two weeks of age in the feather tip samples, resulting in only 58% positivity at six weeks of age, although from spleen of the same birds still 100% positivity was found. These results, along with the serological results, confirmed that rHVT-ND vaccine virus was taken efficiently by the vaccinated birds and replicated during the whole period tested (six weeks). However, there was a shorter age range when the presence of vaccine virus could be detected in 100% of the feather tip samples compared to the spleen samples.

Weekly serological monitoring of vaccinated birds indicated that antibody response to rHVT-ND was detectable by ELISA with partial sero-positivity starting at two weeks of age (*p* < 0.001 from two weeks of age onwards). From 3 weeks of age 100% positivity was found with increasing antibody level throughout the period tested (Table 3). HI test proved to be less sensitive to detect the onset of antibody response following vaccination: sero-positivity was first detected at 3 weeks of age in 55% of the birds, then with increasing titre 100% positivity was reached by 5 weeks of age (Table 3). HI titres were rather low; the mean titre at termination (at 6 weeks of age) was 4.4 log_2_. The results indicated that rHVT-ND vaccine induced significant antibody response to NDV reasonably fast (2–3 weeks), which could be detected earlier with ELISA (code: NDVS) than with HI test. Nucleoprotein ELISA (code: NDVNP) showed negative results in the non-challenged chickens at six weeks of age: 119 ± 92 and 179 ± 85 titre (mean ± STD) were measured in the vaccinated and the non-vaccinated control group, respectively, that were far below the positivity limit of 993.

Pre-challenge sampling of all vaccinated chickens submitted to challenge showed no significant difference in the antibody titre measured both by ELISA and HI test between the subgroups submitted to intra-muscular or intra-nasal challenge (Appendix A).

### 3.2. Efficacy against NDV Challenge

#### 3.2.1. Prevention of Clinical Signs and Mortality

The non-vaccinated control chickens challenged either at 20 or 28 days of age died within six days following intra-muscular challenges and within 6–7 days following intra-nasal challenges. On the other hand, 100% of vaccinated chickens was clinically protected already at 20 days of age. All vaccinated chickens survived the challenge without showing any clinical signs indicative of ND, regardless the challenge route. Similar results were obtained following challenge at four weeks of age, except that a single chicken showed neurological signs after intra-muscular challenge in the vaccinated group (95% clinical protection). Specificity of the clinical signs was verified for that single bird with the detection of ND specific histological lesions in the brain. Although this single vaccinated chicken was evaluated as non-protected, the time course of the disease (clinical signs started from 8 dpch) and the type of clinical signs (only neurological signs) indicated the presence of a developing immunity to NDV, although with some delay. Results are summarised in Figure 1.

#### 3.2.2. Effect of the Challenge Route on Challenge Virus Shedding by Vaccinated Chickens

Following challenge, non-vaccinated control chickens shed high amount of virus both via the oro-nasal and the cloacal route after both challenge dates. Higher NDV load could be measured in the swabs after intramuscular challenge compared to the amount measured after intra-nasal challenge (Table 4; *p* < 0.001 and *p* = 0.004 for oro-nasal swabs, and *p* = 0.004 and *p* = 0.014 for cloacal swabs when the two control subgroups challenged with different routes were compared after the challenge at 20 and 28 days of age, respectively). The range of mean Ct values obtained in the controls after intramuscular challenge was 14.4–17.5 and 14.2–19.0 in the oro-nasal swabs and 16.6–22.5 and 16.6–23.9 in the cloacal swabs at 20 and 28 days of age, respectively. After the intranasal challenge, Ct values ranged 18.4–22.6 and 16.8–20.3 in oro-nasal swabs and 19.3–26.2 and 19.4–27.0 in cloacal swabs at 20 and 28 days of age, respectively.

In the vaccinated birds, regardless of the challenge route, there was a very strong suppression of the challenge virus shedding after both challenge dates. Following challenge at 20 days of age cloacal shedding was not detectable in the majority of vaccinated chickens (8 out of 10 chickens were negative in both groups) and was at low level in the remaining two birds. The effect of the challenge route on the cloacal shedding was negligible (*p* = 0.87). Regarding oro-nasal shedding there was remarkable difference between the vaccinated groups challenged by the two different routes (*p* = 0.002). While oro-nasal swabs contained no detectable amount of NDV in the great majority of chickens after intramuscular challenge (9 out of 10 chickens were negative), significant amount of NDV could be measured after intra-nasal challenge, when all samples were positive with Ct values ranging between 26.2 and 34.9 Ct (Table 4).

After the challenge at four weeks of age, cloacal shedding was not detectable in any of the vaccinated chickens regardless of the challenge routes. On the other hand oro-nasal shedding showed similar pattern to the one observed after the challenge at 20 days of age: no detectable amount of NDV in the great majority of chickens after intramuscular challenge (9 out of 10 chickens were negative), while significant amount of the challenge virus could be detected from the birds challenged by the intra-nasal route (Ct values ranging between 25.2 and 32.0 Ct), with the exception of a single chicken, in which no NDV could be detected (Table 4). The effect of the challenge route on the oro-nasal shedding was strongly significant (*p* < 0.001).

#### 3.2.3. Humoral Immune Response to Challenge

Immune response to challenge could be evaluated in the vaccinated groups only since all non-vaccinated chickens succumbed to challenge. Results at the end of post-challenge observation period were compared to the results obtained at the same age in the vaccinated non-challenged chickens. There was a slight, but significant increase in both HI and ELISA (code: NDVS) titres (*p* ≤ 0.002 for all comparisons; Figure 2), indicating anamnestic response to the challenge infection, regardless the route of the challenge and age at challenge. At the same time nucleoprotein ELISA detected immune response to challenge infection only in a few vaccinated birds (10–15%) after the challenge at four weeks of age (Table 5).

### 3.3. Comparison of the Challenge Virus F Gene Sequence with F Gene Insert of rHVT-ND

Phylogenetic analysis using the phylogenetic tree of Dimitrov et al. for typing of new strains indicated that the challenge strain (NDV-B7-RLQP-CH-EG-12 strain, ref. D3538 in the phylogenetic tree) belongs to sub-genotype VII.1.1 (former genotype VIId, [2]) and clusters together with recent field isolates from Asia, Middle East and North Africa (Appendix A). The inserted F gene in rHVT-ND vaccine (originated from strain D26 of genotype I) shows only 86.1% nucleotide sequence homology to the F gene of the challenge virus.

## 4. Discussion

In several regions of the world poultry production continues to suffer significant economic losses due to Newcastle disease. Outbreaks have been attributed to several causes including the lack of biosecurity, inadequate vaccines and vaccination programs, antigenic variation among the field viruses, interference of maternal antibodies with live and killed vaccines and short duration of the immunity [13,32,33,34]. Unfortunately, biosecurity and vaccinations alone have not been sufficient to stop the circulation of virulent NDV strains [13,15]. Although live attenuated and killed ND vaccines have been used for many decades, these vaccines do not completely prevent infection and consequent losses due to mortality and the poor production performance of vaccinated flocks. Furthermore, these flocks will shed substantial amount of virus and contaminate the environment by which the continuous presence and spread of the disease is maintained [13]. Therefore, there has been a need to develop alternative vaccination strategies that would provide stronger and longer lasting immunity even after single vaccine application in the hatchery. Vaccines used to control poultry diseases ideally should be easily administered, safe without inducing side effects or reverting to virulence, and be capable of inducing fast, long-lasting immunity even when applied in face of MDA. Among the possible strategies to achieve some of these desired characteristics, one of the most promising is the use of recombinant HVT-vectored vaccines. Recombinant HVT expressing the F protein of NDV can easily be administered *in ovo* or at 1 day of age without inducing side effects, such as respiratory distress, associated with live ND vaccines. It overcomes maternal derived antibodies because it is cell associated, and it confers life-long protection because it establishes latency and periodically reactivates [24,25,26,28].

The primary objective of ND vaccination is to prevent clinical disease and mortality; however, the decrease in virulent NDV amount shed into the environment is an additional important benefit. ND vaccines available today do not provide sterilizing immunity, even well-vaccinated birds can become infected without clinical signs and shed the virus [13,15,33]. However, the level of viral shedding can be reduced very significantly depending on the characteristics of vaccine used and the quality of vaccination. Part of the difficulty in preventing viral replication and shedding following infection may be attributed to antigenic differences between the field viruses causing the outbreaks and the vaccine viruses [12,34]. The NDV strains employed most commonly as seed strains for live and inactivated vaccines (LaSota, Hitchner B1, Ulster, VG/GA, etc.,) were originally isolated 30 to 60 years ago and are classified within class II as genotypes II or I [13]. There are controversial opinions regarding the continued use of these vaccines, despite the diversity of virulent genotypes circulating worldwide. Some of the scientists arguing that because of the differences in antigenic properties of the F and HN proteins, there is a need to increase the efficacy of vaccines by matching the vaccine and the field virus [13,15,34]. Others still believe that because all NDV strains are grouped into one serotype, a vaccine made from any strain or genotype is capable of inducing immunity to prevent clinical signs and mortality against a challenge with vNDV, and the differences among vaccines regarding their efficacy to control virus shedding are mainly due to the quality of the vaccine and vaccine application [35].

In recent years, sub-genotype VII.1.1 (former VIId) viruses are the most prevalent vNDV in many parts of the world [3,4,5,8,9]. This prompted us to evaluate the efficacy of rHVT-ND vaccine for its potential to provide early protection against the clinical disease and to control virus shedding following challenge with a representative of this genetic group. The results of the presented study demonstrated that day-old vaccination with rHVT-ND vaccine was capable to provide complete protection against the challenge with genotype VII.1.1 vNDV already at three weeks of age irrespective of the challenge routes used. Since this vaccine contains the F gene of a genotype I apathogenic NDV strain (D26), the results obtained indicate that this vaccine could induce strong immunity even against a vNDV strain that belongs to a heterologous genotype. This finding is supported by previously published results against sub-genotype VII.2 [26,36] and genotype V [25].

Vaccinated chickens were highly protected against clinical disease and stopped shedding the challenge virus almost completely by the cloacal route. The control of oro-nasal replication of vNDV was less efficient, with an obvious difference between the two challenge routes. When the challenge was done by the intra-nasal route, 90–100% of the birds excreted the challenge virus by the oro-nasal route, while 90% of birds that were challenged via the intramuscular route did not shed any detectable virus. It should, however, be emphasized that the replication of the challenge virus was highly significantly suppressed by the vaccination both in the intramuscularly and the intra-nasally challenged birds. The systemic immune response induced by the rHVT-ND vaccine—that replicates primarily in lymphoid cells—resulted in a strong suppression of the challenge virus dissemination in the body (as indicated by the lack of cloacal shedding in the majority of vaccinated birds), while the suppression of the challenge virus replication on the upper respiratory mucosal surface although was highly significant (*p* < 0.001), but less efficient. It is assumed that after intra-nasal challenge, when the virus readily reaches the oro-nasal mucosa in a high dose, it could replicate more efficiently in the upper respiratory mucosa than after intramuscular challenge, when the virus has already been controlled by the immune system during its dissemination to reach the mucosa. This very strong suppression of the challenge virus replication is also reflected by the post-challenge serological results. Nucleoprotein ELISA (code: NDVNP)—which selectively measures antibody response to infection after rHVT-ND vaccination—detected only a very weak primary immune response to challenge (10% and 15% sero-positives and low titres in the intra-muscularly and intra-nasally infected group, respectively). Anamnestic response was significant, comparable between the two challenge routes as measured with HI test and ELISA (code: NDVS; Figure 2A). Although rHVT-ND, which expresses the F protein of NDV exclusively, does not elicit an antibody response to the HN protein, the high amount of anti-F antibodies can inhibit the haemagglutination due to steric hindrance [25], therefore the HI test measures partly the primary response to HN protein and also the secondary response to F protein in the post-challenge serum samples.

ND vaccines generally do not prevent vaccinated animals from becoming infected with a vNDV and subsequent shedding of the virus, however, there are significant differences among the vaccines how much they can decrease the amount of virus shed by vaccinated birds [15,33]. The results of the study presented here showed that rHVT-ND vaccine could very efficiently reduce the amount of virus shed (approx. 3 and 6 log_10_ reduction in mean RNA load in oro-nasal swabs after intranasal and intramuscular challenge, respectively; and approximately 4–5 log_10_ reduction in mean RNA load in cloacal swabs) or could even stop it, which could result in preventing the spread and circulation of vNDV in vaccinated flocks.

## 5. Conclusions

In summary the results of the studies presented in this paper support and extend previous findings regarding the efficacy of rHVT-ND vaccine in preventing the development of clinical signs, and suppressing very strongly, the challenge virus replication and shedding following infection with a genetically (antigenically?) heterologous vNDV strain. By applying this vaccine at the hatchery, controlled vaccine uptake and an efficient homogenous level of immunity can be provided to the vaccinated flocks.

## Figures and Tables

**Figure 1 vaccines-09-00037-f001:**
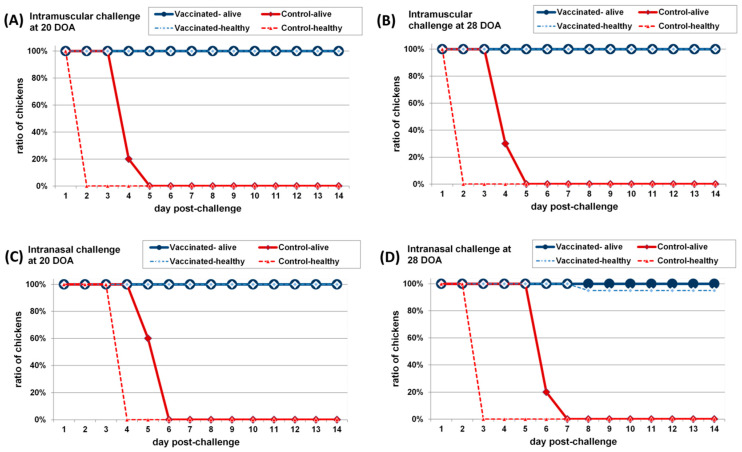
Time course of disease and mortality in vaccinated and non-vaccinated SPF chickens challenged with velogenic Newcastle disease virus. (**A**) intramuscular challenge at 20 days of age; (**B**) intramuscular challenge at 28 days of age; (**C**) intranasal challenge at 20 days of age; (**D**) intranasal challenge at 28 days of age. Ratio of surviving chickens is shown with solid lines, ratio of healthy chickens, without clinical signs indicative to velogenic NDV infection is shown with dotted lines in each graph.

**Figure 2 vaccines-09-00037-f002:**
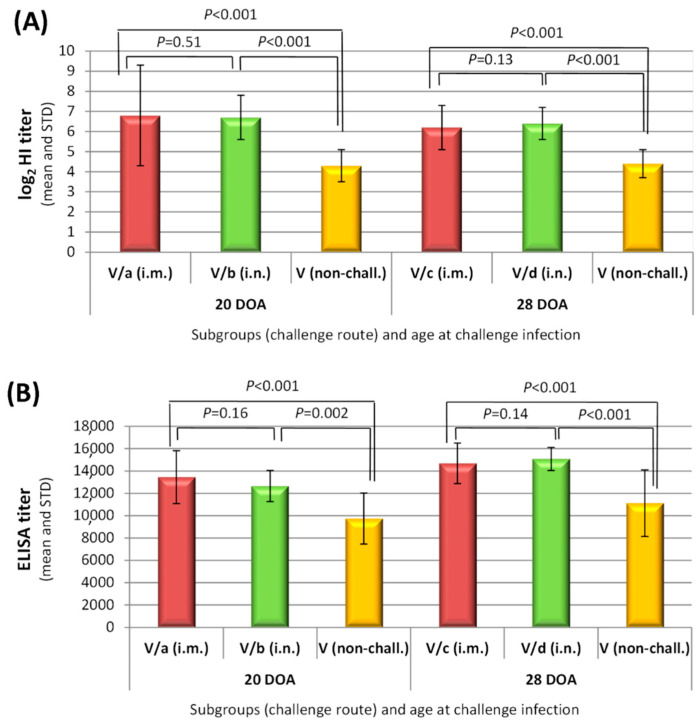
Booster effect of the challenge infection on the humoral immune response to vaccination as measured 14 days post-challenge. Hemagglutination Inhibition test (**A**) and IDScreen^®^ ND indirect ELISA (**B**) results. Day-old SPF chickens were vaccinated subcutaneously and separate subgroups were submitted to velogenic NDV challenge, either by intra-muscular or intra-nasal route at 20 or 28 days of age. Mann–Whitney test was used for the comparison of post-challenge serological results with the corresponding non-challenged vaccinated group at the same age (“effect of challenge”) and for the comparison of corresponding intra-muscularly and intra-nasally challenged groups at the same age (“effect of challenge route”). Results are shown above the columns. DOA: days of age.

**Table 1 vaccines-09-00037-t001:** Study design.

Groups	Subgroups	Age at Challenge	Route of Challenge ^b^	Number of Birds
**Vaccinated ^a^**	V/a	20 DOA ^c^	Intra-muscular	20
	V/b	20 DOA	Intra-nasal	20
	V/c	28 DOA	Intra-muscular	20
	V/d	28 DOA	Intra-nasal	20
	V/Nch ^d^	-	-	12
**Control**	C/a	20 DOA	Intra-muscular	10
	C/b	20 DOA	Intra-nasal	10
	C/c	28 DOA	Intra-muscular	10
	C/d	28 DOA	Intra-nasal	10

^a^ Chicks were vaccinated with 1 dose of rHVT-ND vaccine on the day of hatch subcutaneously. ^b^ Chickens were challenged with a dose of 5.0 log_10_ ELD_50_ regardless the route. One subgroup of the vaccinated group was kept unchallenged until the termination of the second challenge. ^c^ DOA: days of age. ^d^ Nch: non challenged.

**Table 2 vaccines-09-00037-t002:** Detection of vaccine-virus in feather tip pulps or spleen samples in the vaccinated group.

Age of Chickens (Day)	7	15	20		36	43	
Sample type	Feather pulp	Feather pulp	spleen	Feather pulp	spleen	spleen	Feather pulp
**Positivity ^a^**	**90%**	**100%**	**100%**	**100%**	**100%**	**100%**	**58%**
Positivity (positives/tested)	18/20	20/20	10/10	20/20	20/20	12/12	7/12
Ct value mean ± STD (rHVT-ND)	31.0 ± 3.8	24.8 ± 3.8	29.0 ± 0.9	29.7 ± 5.6	30.6 ± 1.9	31.6 ± 3.3	36.8 ± 3.6
Ct value mean ± STD (ovotransferrin)	23.1 ± 0.7	23.2 ± 0.9	18.9 ± 0.5	23.5 ± 0.9	18.9 ± 0.9	18.8 ± 0.6	24.0 ± 1.5
ΔCt ^b^	7.9 ± 3.6	1.6 ± 3.8	10.1 ± 1.2	6.2 ± 5.8	11.7 ± 1.9	12.9 ± 3.2	12.8 ± 3.7

^a^ all samples with Ct value below 40 was considered positive, ^b^ relative quantification of rHVT-ND (ΔCt = Ct _rHVT-ND_ − Ct_ovotransferrin_).

**Table 3 vaccines-09-00037-t003:** Humoral immune response to vaccination- ELISA and hemagglutination-inhibition test results.

Method	Age at Sampling (Day)		7	15	20	28	35	43
	Parameter	Group						
**IDScreen^®^ ND Indirect ELISA**	ELISA titer mean ± STD	Vaccinated	31 ± 39	2516 ± 1936	3964 ± 1697	8608 ± 2215	9743 ± 2294	11,119 ± 2976
		Control	NS	NS	137 ± 287	55 ± 105	86 ± 184	1 ± 0
	Positivity	Vaccinated	0%	80%	100%	100%	100%	100%
		Control	NS	NS	0%	0%	0%	0%
**HI test**	Log_2_ HI titer mean ± STD	Vaccinated	0.1 ± 0.2	0.9 ± 0.5	1.6 ± 0.9	2.5 ± 0.7	4.3 ± 0.8	4.4 ± 0.7
		Control	NS	NS	0.0 ± 0.0	0.2 ± 0.4	1.9 ± 0.2	1.7 ± 0.4
	Positivity	Vaccinated	0%	0%	55%	90%	100%	100%
		Control	NS	NS	0%	0%	0%	0%

SPF chickens were vaccinated at day-old with rHVT-ND vaccine subcutaneously. Representative numbers of vaccinated non-challenged chickens were tested weekly to follow the development of humoral immune response to vaccination (*n* = 20 at 7, 15 and 20 days of age, *n* = 40 at 28 days of age and *n* = 12 at 35 and 43 days of age). In the non-vaccinated group, sero-negativity was verified from the first challenge date onwards (*n* = 7–10 at each sampling). Positivity limit of ELISA (code: NDVS) is above 993, positivity limit of haemagglutination inhibition (HI) test is at least 2 log_2_ HI titre. NS no sample collected.

**Table 4 vaccines-09-00037-t004:** Challenge virus shedding (measured with one-step RT-real-time PCR).

Age at Challenge	Route of Challenge	Sub-Group	Oro-Nasal Swabs			Cloacal Swabs		
			Ct (Mean ± STD and Range)	*p*-Value ^§^	Positivity ^#^	Ct (Mean ± STD and Range)	*p*-Value ^§^	Positivity ^#^
20 DOA *	i.m.	V/a (vaccinated)	35.0 ± 3.2 (25.8–36.0)	*p* < 0.001	1/10	34.4 ± 3.4 (26.7–36.0)	*p* < 0.001	2/10
		C/a (control)	16.2 ± 0.9 (14.4–17.5)		10/10	19.4 ± 1.6 (16.6–22.5)		10/10
	i.n.	V/b (vaccinated)	29.8 ± 3.2 (26.2–34.9)	*p* < 0.001	10/10	35.4 ± 1.3 (32.4–36.0)	*p* < 0.001	2/10
		C/b (control)	19.7 ± 1.4 (18.4–22.6)		10/10	22.3 ± 2.0 (19.3–26.2)		10/10
28 DOA	i.m.	V/c (vaccinated)	35.8 ± 0.7 (33.8–36.0)	*p* < 0.001	1/10	36.0 ± 0.0 (36.0–36.0)	*p* < 0.001	0/10
		C/c (control)	16.7 ± 1.6 (14.2–19.0)		10/10	21.0 ± 2.6 (16.6–23.9)		10/10
	i.n.	V/d (vaccinated)	30.1 ± 4.1 (25.2–36.0)	*p* < 0.001	9/10	36.0 ± 0.0 (36.0–36.0)	*p* < 0.001	0/10
		C/d (control)	18.8 ± 1.0 (16.8–20.3)		10/10	23.9 ± 2.2 (19.4–27.0)		10/10

* DOA: days of age, ^#^ Positivity limit: Ct below 36, ^§^ Mann-Whitney test result for comparison of Ct values of vaccinated and corresponding non-vaccinated control groups.

**Table 5 vaccines-09-00037-t005:** Humoral immune response of vaccinated chickens to challenge—Nucleoprotein ELISA results.

Subgroup	Non-Challenged	Intra-Muscularly Challenged	Intra-Nasally Challenged
ELISA titre (mean ± STD)	119 ± 92	621 ± 872	590 ± 389
Positivity	0%	10%	15%

Nucleoprotein-based ELISA (code: NDVNP) was used for the selective measurement of primer immune response to challenge infection. SPF chickens were vaccinated with a rHVT-ND vaccine at day-old, challenged with velogenic NDV at 28 days of age and tested at the end of 14 days long post-challenge observation period. Corresponding non-challenged vaccinated chickens were sampled at the same age. Positivity limit of the test is 993 titre or above.

## Data Availability

The data presented in this study are available on request from the corresponding author.

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
