# Peer review of "Efficacy of a Turkey Herpesvirus Vectored Newcastle Disease Vaccine against Genotype VII.1.1 Virus: Challenge Route Affects Shedding Pattern"

_vaccines, 2021, doi:10.3390/vaccines9010037_

Round 1

Reviewer 1 Report

One of the most common viral diseases, which can cause significant economic problem, is caused by Newcastle disease virus (NDV). Palya et al. tested if herpesvirus of turkeys (HVT) expressing F protein of NDV can be used for vaccination of chickens to protect them from getting infected by NDV. The results show that regardless if the vaccination was given either via the intra-muscular or the intra-nasal route, the chickens were protected against clinical symptoms of NDV infection. However, while cloacal NDV shedding was inhibited in both intra-nasal and intra-muscular vaccinated animals, oro-nasal NDV shedding was higher in the intra-nasal vaccinated groups compared to the intra-muscular vaccinated groups indicating differential efficacy in developing immune control of NDV based on the vaccination route. The study is important for providing important information to designing and developing new vaccines and vaccination protocols to protect poultry against NDV infection. I have only a few comments, which can help to improve the manuscript for the readers.

  1. HVT needs to be spelled out what it means in the abstract. There are a number of acronyms used in the text, figure legends, and tables. All acronyms should be spelled out where they are first mentioned.

  1. The importance of F protein in vaccination/NDV biology should be mentioned in the introduction since the vaccine is based on this viral protein.

  1. Fig 1 shows the phylogenetic analysis of the NDV strain used in this study. What is so special about this phylogenetic tree of NDVs based on their F gene compared to previous phylogenetic trees? If there is a difference, it should be discussed in the Results section and not just mentioned in the Methods section. Otherwise, I do not see why it is displayed as Fig 1.

  1. Line 190. The qPCR detection of HVT-ND does not show the replication of the vaccine virus. Rather, it detects the presence of HVT-ND. The statement needs to be revised.

  1. Table 2, the experiment description related to it and their interpretation are not clearly explained. Are there any differences in HVT-ND detection between intra-muscular and intra-nasal challenged chicks? Why was group d not included in the analysis? It would be better to present both the qPCR and the serological data over time by intra-nasal and intra-muscular challenged groups.

Is it possible that the increased Ct value of HVT-ND DNA in qPCR at d15 relative to d5 in feather pupl indicates initial viral replication and then establishment of viral latency from d15 (displaying reduced viral DNA detection from this time point) given that a herpesvirus was used as a vaccine vector?

  1. Line 231. “At both ages, controls died within six days…” What ages are we talking about here? While it was mentioned in the Methods section, it would help the reader if it were mentioned here as well.

  1. Also, it is unclear how exactly the vaccination/NDV challenge was performed. This should be described in 3.2.1 before the results are presented.

  1. Paragraph 3.3 should be at the beginning of the Results section and not at the end.

Author Response

Dear Reviewer,

Thank you for considering our manuscript “Efficacy of a Turkey Herpesvirus Vectored Newcastle Disease Vaccine against Genotype VII.1.1 Challenge Virus: Challenge Route Affects Shedding Pattern’ for publication in Vaccines. We are grateful for the opportunity to resubmit a revised manuscript, which was amended according to the Reviewers’ requests and suggestions. You will find our response (in blue) to your comments and questions (in black) point-by-point. Modifications in the submitted manuscript have been done using the "Track Changes" function in Microsoft Word.

The authors would like to thank for the thorough review of our article. Your comments and recommendations are highly appreciated by the authors.

Best regards,

The authors

One of the most common viral diseases, which can cause significant economic problem, is caused by Newcastle disease virus (NDV). Palya et al. tested if herpesvirus of turkeys (HVT) expressing F protein of NDV can be used for vaccination of chickens to protect them from getting infected by NDV. The results show that regardless if the vaccination was given either via the intra-muscular or the intra-nasal route, the chickens were protected against clinical symptoms of NDV infection. However, while cloacal NDV shedding was inhibited in both intra-nasal and intra-muscular vaccinated animals, oro-nasal NDV shedding was higher in the intra-nasal vaccinated groups compared to the intra-muscular vaccinated groups indicating differential efficacy in developing immune control of NDV based on the vaccination route. The study is important for providing important information to designing and developing new vaccines and vaccination protocols to protect poultry against NDV infection. I have only a few comments, which can help to improve the manuscript for the readers.

Before start responding to your questions and comments we would like to clarify a confusion, that can be due to typing error, regarding the route of vaccine application and the route of challenge infection we used in our study. As we described it in the M&M section we used only one route of vaccine application, the subcutaneous route, to immunize the birds, while the challenges were done by two different routes (i.m. or oro-nasal). We designed our study this way to generate information on the strength of immunity (systemic versus mucosal) induced by the rHVT-ND vaccine.

Regarding your specific comments/questions please find below our answers.

1. HVT needs to be spelled out what it means in the abstract. There are a number of acronyms used in the text, figure legends, and tables. All acronyms should be spelled out where they are first mentioned.

 Done. We did review the paper and did corrections according to the request.

2. The importance of F protein in vaccination/NDV biology should be mentioned in the introduction since the vaccine is based on this viral protein.

Amendment of the introduction section has been done as suggested.

3. Fig 1 shows the phylogenetic analysis of the NDV strain used in this study. What is so special about this phylogenetic tree of NDVs based on their F gene compared to previous phylogenetic trees? If there is a difference, it should be discussed in the Results section and not just mentioned in the Methods section. Otherwise, I do not see why it is displayed as Fig 1.

We included the phylogenetic tree to show the genetic distance of vaccine insert from the challenge virus and also to demonstrate the relatedness of the challenge virus to other field isolates. We agree with the proposal to delete it from the results section and we moved it to a supplementary file.

4. Line 190. The qPCR detection of HVT-ND does not show the replication of the vaccine virus. Rather, it detects the presence of HVT-ND. The statement needs to be revised.

 Thanks for your clarification. We did the modification in the text according to your comment.

5. Table 2, the experiment description related to it and their interpretation are not clearly explained. Are there any differences in HVT-ND detection between intra-muscular and intra-nasal challenged chicks? Why was group d not included in the analysis? It would be better to present both the qPCR and the serological data over time by intra-nasal and intra-muscular challenged groups.

Is it possible that the increased Ct value of HVT-ND DNA in qPCR at d15 relative to d5 in feather pupl indicates initial viral replication and then establishment of viral latency from d15 (displaying reduced viral DNA detection from this time point) given that a herpesvirus was used as a vaccine vector?

We aimed to reduce the number of chickens used in the study according to the 3R concept for animal welfare. This is why we used two separate sample types (feather and spleen) for rHVT-ND detection and optimized the samplings with the inclusion of reduced number of non-challenged and challenged chickens as well. The aim of these testings was to verify the proper vaccine administration through the detection of rHVT-ND presence in the spleen and its replication in the feathers. We considered that each sampling date represents the whole vaccinated group of chickens. Therefore, we skipped giving all details in the footer of the table in the revised manuscript. This makes it easier to follow.

The aim of rHVT-ND detection was only to support the proper vaccine administration on a representative number of chickens and compare the two possible sample types for the suitability of the detection of vaccine-take. It was not the objective of the study to use this parameter for the comparison of subgroups challenged by the different routes.

The birds forming the different subgroups were assigned randomly only on the day of the challenge without knowing their serological status, therefore the only possibility was to make comparison of serological results obtained on the day of the challenge. The SPF chickens were vaccinated individually by subcutaneous injection which results in homogenous vaccine-take. All subgroups were kept together before the challenge, therefore no difference would be expected between them. This is supported by the pre-challenge serological results of corresponding intra-nasally and intra-muscularly challenged subgroups. The comparative analysis of serological results by subgroups was added to the revised manuscript as supplementary table 1.

6. Line 231. “At both ages, controls died within six days…” What ages are we talking about here? While it was mentioned in the Methods section, it would help the reader if it were mentioned here as well.

 Modification of the text has been done as suggested.

7. Also, it is unclear how exactly the vaccination/NDV challenge was performed. This should be described in 3.2.1 before the results are presented.

In the M&M section the method and route of vaccination are described in details. The routes of challenge and the applied challenge virus dose are also described in the M&M section and repeated in the results section. We did not think it would be necessary to repeat how the vaccination was done since it was similar for all groups of vaccinated chickens.

8. Paragraph 3.3 should be at the beginning of the Results section and not at the end.

Since this para provides the serological results obtained after the challenge infection we believe it logically better fits after the para describing the challenge results.

Reviewer 2 Report

Palya et al report analysis of vaccination of chickens against Newcastle Disease using a recombinant HVT vaccine. Complete clinical protection was observed and cloacal and oro-nasal shedding of challenge virus was monitored.

The study is interesting, data are well presented and the authors' conclusions justified by the data. I have no major issues with this study.

Minor comments:

  1. The English language would benefit from editing in places.
  2. Fig. 1 is largely illegible - this should be enlarged. However, inclusion may not be absolutely necessary.
  3. Tables 2 and 4 would be more easily interpretable if Ct values were recalculated as RNA concentrations.

Author Response

Dear Reviewer,

Thank you for considering our manuscript “Efficacy of a Turkey Herpesvirus Vectored Newcastle Disease Vaccine against Genotype VII.1.1 Challenge Virus: Challenge Route Affects Shedding Pattern’ for publication in Vaccines. We are grateful for the opportunity to resubmit a revised manuscript, which was amended according to the Reviewers’ requests and suggestions. You will find our response (in blue) to your comments and questions (in black) point-by-point. Modifications in the submitted manuscript have been done using the "Track Changes" function in Microsoft Word.

The authors would like to thank for the thorough review of our article. Your comments and recommendations are highly appreciated by the authors.

Best regards,

The authors

Palya et al report analysis of vaccination of chickens against Newcastle Disease using a recombinant HVT vaccine. Complete clinical protection was observed and cloacal and oro-nasal shedding of challenge virus was monitored.

The study is interesting, data are well presented and the authors' conclusions justified by the data. I have no major issues with this study.

Minor comments:

1. The English language would benefit from editing in places.

English language editing was done throughout the article.

2. Fig. 1 is largely illegible - this should be enlarged. However, inclusion may not be absolutely necessary.

Fig.1 was removed from the results section and moved to Supplementary material. The resolution of the image allows proper magnification in the electronic format.

3. Tables 2 and 4 would be more easily interpretable if Ct values were recalculated as RNA concentrations.

In both measurements a fixed threshold was established using certain dilutions of the challenge virus stock (for NDV quantification) and the vaccine (for rHVT-ND quantification), therefore the Ct values even if they were obtained in a separate run are comparable. Efficacy of the RT-qPCR reaction for NDV quantification was 0.99, therefore 1 Ct difference can be considered as a two-fold difference in NDV RNA copy number.